# The CovR regulatory network drives the evolution of Group B *Streptococcus* virulence

**Maria-Vittoria Mazzuoli**[1,2], **Maëlle Daunesse**[3¤], **Hugo Varet**[3,4], **Isabelle Rosinski-Chupin**[5], **Rachel Legendre**[3,4], **Odile Sismeiro**[1,4], **Myriam Gominet**[1], **Pierre Alexandre Kaminski**[1], **Philippe Glaser**[5], **Claudia Chica**[3], **Patrick Trieu-Cuot**[1], **Arnaud Firon**[1] *

**1** Unité Biologie des Bactéries Pathogènes à Gram-positif, CNRS UMR2001 Microbiologie Intégrative et Moléculaire, Institut Pasteur, Paris, France, **2** Sorbonne Paris Cité, Université de Paris, Paris, France, **3** Hub de Bioinformatique et Biostatistique—Département Biologie Computationnelle, Institut Pasteur, Paris, France, **4** Plate-forme Technologique Biomics—Centre de Ressources et Recherches Technologiques (C2RT), Institut Pasteur, Paris, France, **5** Unité Écologie et Évolution de la Résistance aux Antibiotiques, CNRS UMR3525, Institut Pasteur, Paris, France

¤ Current address: European Molecular Biology Laboratory, European Bioinformatics Institute, Wellcome Genome Campus, Hinxton, Cambridge United Kingdom

\* arnaud.firon@pasteur.fr

**Data Availability Statement:** Sequencing data (RNA-seq, dRNA-seq, ChIP-seq) are publicly available in the GEO database (https://www.ncbi.nlm.nih.gov/geo/query/acc.cgi?acc=GSE158049).

## Abstract

Virulence of the neonatal pathogen Group B *Streptococcus* is under the control of the master regulator CovR. Inactivation of CovR is associated with large-scale transcriptome remodeling and impairs almost every step of the interaction between the pathogen and the host. However, transcriptome analyses suggested a plasticity of the CovR signaling pathway in clinical isolates leading to phenotypic heterogeneity in the bacterial population. In this study, we characterized the CovR regulatory network in a strain representative of the CC-17 hypervirulent lineage responsible of the majority of neonatal meningitis. Transcriptome and genome-wide binding analysis reveal the architecture of the CovR network characterized by the direct repression of a large array of virulence-associated genes and the extent of co-regulation at specific loci. Comparative functional analysis of the signaling network links strain-specificities to the regulation of the pan-genome, including the two specific hypervirulent adhesins and horizontally acquired genes, to mutations in CovR-regulated promoters, and to variability in CovR activation by phosphorylation. This regulatory adaptation occurs at the level of genes, promoters, and of CovR itself, and allows to globally reshape the expression of virulence genes. Overall, our results reveal the direct, coordinated, and strain-specific regulation of virulence genes by the master regulator CovR and suggest that the intra-species evolution of the signaling network is as important as the expression of specific virulence factors in the emergence of clone associated with specific diseases.

## Author summary

*Streptococcus agalactiae*, commonly known as the Group B Streptococcus (GBS), is a commensal bacterium of the intestinal and vaginal tracts found in approximately 30% of

All other data are within the manuscript and its Supporting Information files.

**Funding:** This work was supported by grants from the French Government 'Laboratory of Excellence - Integrative Biology of Emerging Infectious Diseases' (LabEx IBEID, grant number ANR-10-LABX-62-IBEID to PTC), the Fondation pour la Recherche Médicale (FRM grant number DEQ20181039599 to PTC), and the ANR (HemeDetox grant number ANR-17-CE11-0044-03 to AF). The funders had no role in study design, data collection and analysis, decision to publish, or preparation of the manuscript.

**Competing interests:** The authors have declared that no competing interests exist.

healthy adults. However, GBS is also an opportunistic pathogen and the leading cause of neonatal invasive infections. Epidemiologic data have identified a particular GBS clone, designated the CC-17 hypervirulent clonal complex, as responsible for the overwhelming majority of neonatal meningitis. The hypervirulence of CC-17 has been linked to the expression of two specific surface proteins increasing their abilities to cross epithelial and endothelial barriers. In this study, we characterized the role of the major regulator of virulence gene expression, the CovR response regulator, in a representative hypervirulent strain. Transcriptome and genome-wide binding analysis reveal the architecture of the CovR signaling network characterized by the direct repression of a large array of virulence-associated genes, including the specific hypervirulent adhesins. Comparative analysis in a non-CC-17 wild type strain demonstrates a high level of plasticity of the regulatory network, allowing to globally reshape pathogen-host interaction. Overall, our results suggest that the intra-species evolution of the regulatory network is an important factor in the emergence of GBS clones associated with specific pathologies.

## Introduction

*Streptococcus agalactiae*, commonly known as Group B *Streptococcus* (GBS), is the leading cause of sepsis and meningitis in the first three months of life and a significant cause of *in utero* infections and preterm births [1,2]. The reconstitution of the evolutionary history of the species suggests that the human-adapted strains have emerged in the mid-twentieth century, corresponding to the period of the first clinical cases [3]. Of the five main clonal complexes (CCs) associated with human infections, strains of the CC-17 lineage are responsible of the vast majority of late-onset meningitis in neonates and, consequently, are classified as the hypervirulent GBS clones. CC-17 strains are specifically associated with human and are highly homogenous compared to strains belonging to other clonal complexes, a characteristic of an epidemic clone with worldwide dissemination [3–5].

The success of the hypervirulent clone as a neonatal pathogen is linked to the expression of two specific adhesins, HvgA and Srr2 [6]. The HvgA adhesin is an allelic variant of the BibA protein present in non-CC17 strains and confers a higher ability to translocate through different host barriers, most notably through the blood-brain barrier [7]. The Srr2 serine-rich protein, which is mutually exclusive with the Srr1 adhesins expressed by non-CC-17 strains, enhances the capacity to cross the intestinal barrier in the developing neonatal gastrointestinal tract [8,9]. Proteins covalently anchored to the cell-wall by their LPxTG motif through the activity of the sortase A enzyme [10], such as HvgA and Srr2, are a major group of virulence factors with adhesion or immune-modulatory properties [11]. These cell-wall anchored proteins are subject to selective pressure generating variability in the GBS population either through allelic variation (*e.g* the *bibA/ hvgA* alleles) or gain and loss of virulence genes (*e.g* the *srr1/srr2* mutually exclusive loci) or mobile elements [11–13]. In addition, a precise and coordinated control of the expression of virulence genes is essential for GBS pathogenesis. The expression of the appropriate combination of surface proteins and of secreted factors, most notably a ß-hemolytic/cytotoxic toxin (ß-h/c) [14,15], is essential for GBS to establish commensal relationships within the adult vaginal and intestinal tracts as well as to become an extracellular pathogen in susceptible hosts [16–18].

The major regulator of host-pathogen interaction in ß-hemolytic streptococci is the transcriptional factor CovR belonging to the OmpR family of bacterial response regulator [19,20]. Targeted analysis in GBS demonstrated that CovR directly represses the transcription of the

*cyl* operon encoding the ß-h/c toxin, the *bibA* gene, and the PI-1 pili operon [21–24]. The regulation by CovR is highly dynamic and sustains the trade-off between cytotoxicity and adherence, ultimately leading to bacterial multiplication or elimination by the host immune response [14,25,26]. The activity of CovR is modulated by its cognate histidine kinase CovS, which has a dual kinase and phosphatase activity on a conserved $D_{53}$ aspartate residue [27,28]. The dynamic equilibrium between the opposite enzymatic activities depends on the interaction of CovS with the membrane protein Abx1 [27], and on the mutually exclusive phosphorylation of a CovR threonine residue ($T_{65}$) by the serine-threonine kinase Stk1 [24].

In spite of being the major regulator of virulence in GBS, previous transcriptomic analyses of the CovR/S two-component system have been done in non-CC-17 isolates only. The inactivation of CovR is usually associated with a global transcriptome remodelling involving 10 to 15% of the genes, with CovR being mainly a transcriptional repressor. However, important variation in the CovR regulon was observed early on among strains of different clonal complexes [22,26,27]. In addition, a genomic analysis has highlighted mutational biases in the CovR/S system itself and in the CovR regulated promoters in CC-17 strains, suggestive of a positive selection acting on the CovR regulatory pathway in hypervirulent clones [4]. Evolution of regulatory pathways is a common process observed between related bacterial species [29], but our knowledge on intra-species regulatory evolution remains limited [30–32]. To address the role of CovR in GBS hypervirulent clones, we characterized the CovR regulon in a CC-17 strain and demonstrated the direct regulation of a combination of proteins involved in host-pathogen interaction. Comparative analysis revealed strain-specificities supported by mechanisms acting locally on CovR regulated genes and promoters and globally at the level of CovR activation by phosphorylation. The plasticity of the CovR regulatory network generates phenotypic heterogeneity at the species level, thus allowing the selection of clones associated to specific hosts and pathological conditions.

## Results

### CovR regulates virulence genes expression and prophages transcription in BM110

To define the transcriptional response associated to CovR inactivation in the hypervirulent CC-17 lineage, we constructed two *covR* mutants in the BM110 strain [3]. The first mutant has an in-frame deletion of the *covR* sequence (Δ*covR*) and the second a two base-pairs chromosomal substitution (AT->CC) resulting in the translation of a CovR$_{D53A}$ variant that cannot be phosphorylated by CovS. Phenotypically, both mutants are hyper-haemolytic and hyper-pigmented, as observed for *covR* mutants in other backgrounds (S1 Fig). Transcriptome analysis by RNA-seq of the CovR$_{D53A}$ mutant revealed a differential expression for 12.2% of the genes (N = 266/2178; |Log$_2$ FC| > 1; adjusted p-value < 0.005) at mid-exponential growth phase in rich medium (S1A Table). Overall, fold changes associated to the 137 up-regulated genes were higher than those associated to the 129 down-regulated genes (Fig 1A and S1B and S1C Table). Comparison of the CovR$_{D53A}$ and Δ*covR* transcriptomes highlighted a clear correlation for highly differentially expressed genes but also emphasised specificities (Fig 1B and 1C). The most striking difference was the opposite regulation of clusters of genes located in four prophages (S2 Fig). These four prophages accounted for a large proportion of the variability between mutants, encompassing 61.2% (79/129) of the down-regulated genes in the CovR$_{D53A}$ and 34.2% (128/374) of the up-regulated genes in the Δ*covR* mutant (S1 Table).

The transcription of 76 and 3 genes (= 3.6% of the total number of genes) is similarly repressed or activated, respectively, in the two *covR* mutants when excluding the four prophages (S1F Table). Among them, genes highly repressed (FC > 10) encodes for major

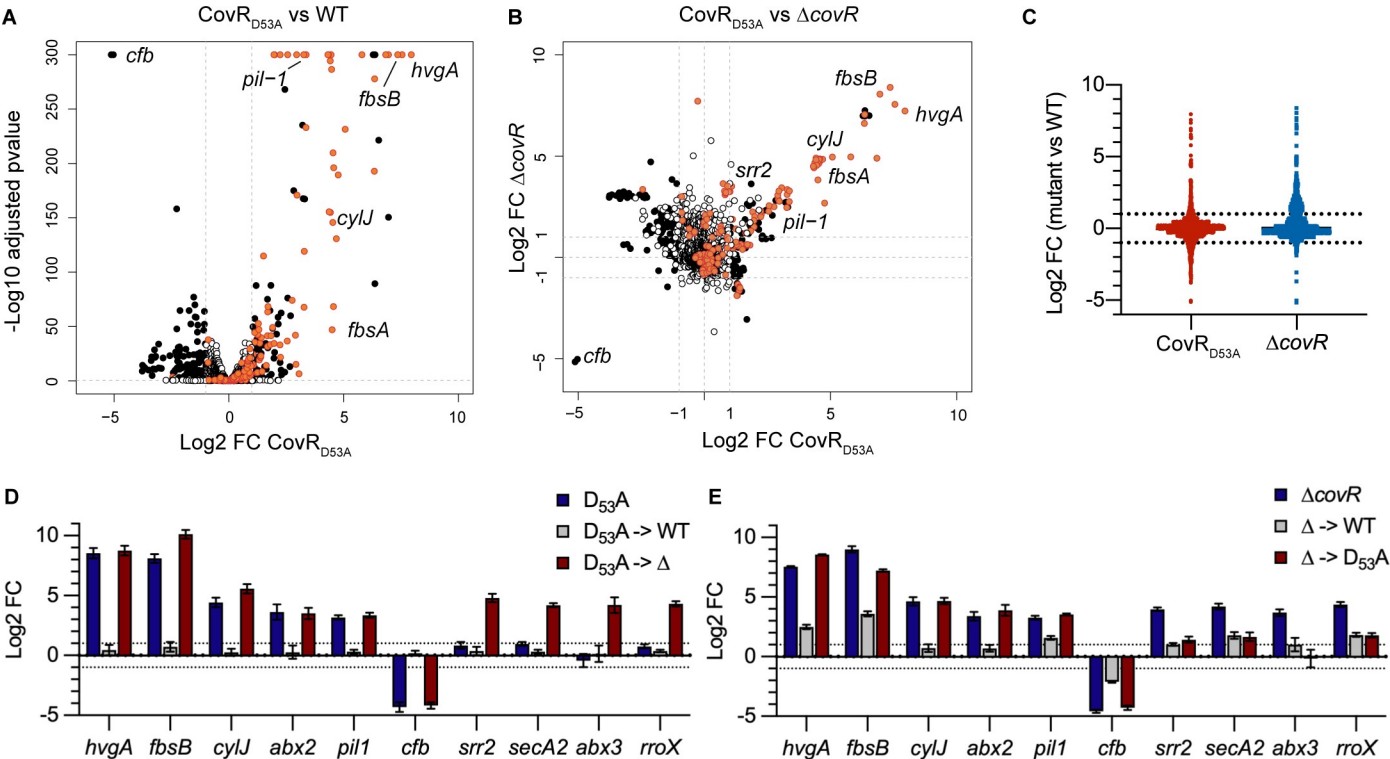

**Fig 1. The CovR regulon in the hypervirulent BM110 strain.** (A) Volcano plot of the BM110 CovR$_{D53A}$ transcriptome at mid-exponential phase in THY. Each dot represents one of the 2178 genes with its RNA-seq fold change and adjusted p-value (Wald test) calculated from three independent replicates. Black and white dots symbolized significant ($|Log_2 FC| > 1$; adjusted p-value < 0.005) and non-significant differentially transcribed genes, respectively. Red dots are genes associated to a CovR binding region identified by ChIP-Seq (see Fig 2), with selected gene names highlighted. (B) Pairwise comparisons of the BM110 CovR$_{D53A}$ and $\Delta covR$ mutant transcriptomes. Each dot has the same color-coded as in (A), corresponding to significant vs non-significant differential gene expression in the CovR$_{D53A}$ mutant. Dots dispersion represents mutant-specificities. (C) Dot plots of RNA-seq fold changes for all genes in the $\Delta covR$ (blue) and CovR$_{D53A}$ (red) mutants. Note the opposite trends in the total number of mild up-regulated genes ($1 < log_2 FC < 3$) in the $\Delta covR$ mutant and of down-regulated genes ($-3 < log_2 FC < -1$) in the CovR$_{D53A}$ mutant. (D) Validation of gene expression by qRT-PCR in the CovR$_{D53A}$ mutant (D$_{53}$A), the chromosomally complemented strain (D$_{53}$A->WT), and in a $\Delta covR$ mutant done in the CovR$_{D53A}$ background (D$_{53}$A->$\Delta$). (E) Validation of gene expression by qRT-PCR in the $\Delta covR$ mutant ($\Delta$), the chromosomally complemented strain ($\Delta$->WT), and in a CovR$_{D53A}$ mutant done in the $\Delta covR$ background ($\Delta$->D$_{53}$A). Means and standard deviations of log2 fold change (mutant versus WT) are calculated from three biological replicates.

virulence factors such as the CC-17-specific adhesin HvgA [7], the LPxTG adhesin FbsA [33], the PI-1 pili operon and its associated regulator [23,34], and the *cyl* operon necessary for the synthesis and export of the ß-h/c toxin [14] (Fig 1B and S1F Table). In addition, CovR strongly repressed (10 < FC < 234) seven genes encoding secreted proteins (S1F Table), among which the FbsB adhesin [35] and the NucA endonuclease [36], as well as an overlooked set of small proteins (51 to 157 residues after signal peptide cleavage). The regulation of a large combination of cell-wall and secreted proteins establishes CovR as the central regulator of host-pathogen interaction in CC-17.

In addition to the 79 similarly regulated genes, 7 and 64 genes are differentially transcribed more than four times ($|Log_2 FC| > 2$) only in the CovR$_{D53A}$ or $\Delta covR$ mutants, respectively (S1 Table). Most notably, the operon encoding for the CC-17-specific adhesin Srr2 is differentially expressed only in the $\Delta covR$ mutant (Fig 1B). Quantitative RT-PCRs on independent cultures corroborate the RNA-seq fold changes in the two mutants for 10 selected genes (Fig 1D and 1E). Re-introduction of a WT *covR* allele at its chromosomal locus restores WT or near WT level of transcription in the CovR$_{D53A}$ (Fig 1D) and $\Delta covR$ (Fig 1E) mutants, respectively. To independently test the specificities of the *covR* mutations, we swapped the mutations by

constructing a CovR$_{D53A}$ mutant in the $\Delta covR$ background ($\Delta$->D$_{53}$A mutant) and, recipro-cally, a $\Delta covR$ mutant in the CovR$_{D53A}$ background (D$_{53}$A->$\Delta$ mutant). For the four genes selected based on their RNA-seq overexpression only in the $\Delta covR$ mutant, RT-qPCR with the swapped mutants confirmed their differential expression when *covR* is deleted but not when CovR is inactivated by a D$_{53}$A point mutation (Fig 1D and 1E). The difference between the two mutants suggests indirect effects in the $\Delta covR$ mutant, which might be due to a crosstalk activity of CovS in the absence of its cognate regulator [37,38], or alternatively, to the binding of the non-phosphorylated CovR$_{D53A}$ variant on specific promoters, as has been suggested for the CovR orthologue of *S. pyogenes* [28,39–41] and more generally for non-phosphorylated form of response regulators [42].

## CovR binds to promoter regions along the BM110 chromosome

To identify genes directly regulated by CovR, we ectopically expressed a N-terminal epitope-tagged CovR variant (FLAG-CovR) in the BM110 $\Delta covR$ mutant. Expression of FLAG-CovR depends on the addition of anhydro-tetracycline (aTc) and led to a dose-dependent repression of selected genes up-regulated in the parental $\Delta covR$ mutant (S1 Fig). The dose-dependent transcriptional repressions are similar to the levels observed with the ectopic expression of a WT *covR* allele, indicating that the FLAG epitope does not significantly impact the functional-ity of CovR (S1 Fig). Therefore, the FLAG-CovR variant was used for ChIP-sequencing with two independent cultures of exponentially growing bacteria in THY after induction with 50 ng/ml aTc, altogether with a similar non-induced condition (no-aTc) and an additional strain with a non-epitope tagged CovR (no-TAG) as controls.

After sequencing, analysis and manual curation, we detected 62 high-confidence reproduc-ible loci with sequence enrichment (mean fold enrichment FE > 4, IDR < 0.05) distributed along the 2.17 Mb BM110 chromosome (Fig 2A and S2A Table). At these loci, the distribution of sequencing reads mapped on the chromosome have the typical characteristics of a ChIP-seq signal, forming a peak centered on or near the regulator binding site. The summit of 42 peaks (68%) is localized between -200 and + 100 bp of a start codon (Fig 2B), as expected for a tran-scriptional regulator. To improve the association between the binding peaks and promoters, we mapped all transcriptional start sites (TSSs) by differential RNA sequencing (dRNA-seq). In total, 1,035 TSSs were detected, including 60 associated with small or non-coding RNAs (32 intergenic and 28 antisense RNAs) and 113 TSSs inside ORFs (S3 Table). Genome-wide TSS comparison identified 953 TSSs (92%) conserved between BM110 and the reference strain NEM316 [43], increasing mapping confidence (S3 Table). CovR binding peaks were more closely associated to TSSs than to start codon (Fig 2C), with 39 peak summits (62%) located at less than 100 bp of a TSS (S2A Table).

To confirm CovR binding, we selected eight promoters for *in vitro* electrophoretic mobility shift assay (EMSA). The purified recombinant protein (rCovR) binds to the eight promoters and *in vitro* phosphorylation of rCovR by acetyl-phosphate increased its affinity for all tested promoters (Fig 3). The recombinant rCovR does not bind to the *gyrA* promoter used as a nega-tive control and *in vitro* phosphorylation of rCovR was confirmed by Phos-Tag analysis (S3 Fig). *In vitro* DNaseI protection assay (footprinting) on three DNA loci revealed one or two CovR-protected regions of different lengths by promoters (S3C Fig). Variability in number and length of CovR binding sites was previously observed, suggesting a complex motif archi-tecture [21,24]. Indeed, a clear consensus binding motif was not detected by considering all chromosomal CovR binding loci. The most significant enriched motif is a widespread AT-rich sequence (ATTA(A/G)A(A/T)) present in 54 out of the 62 peaks, which is also the most signifi-cant motif detected by considering only peaks closely associated to a TSS (Fig 2D). This A/T

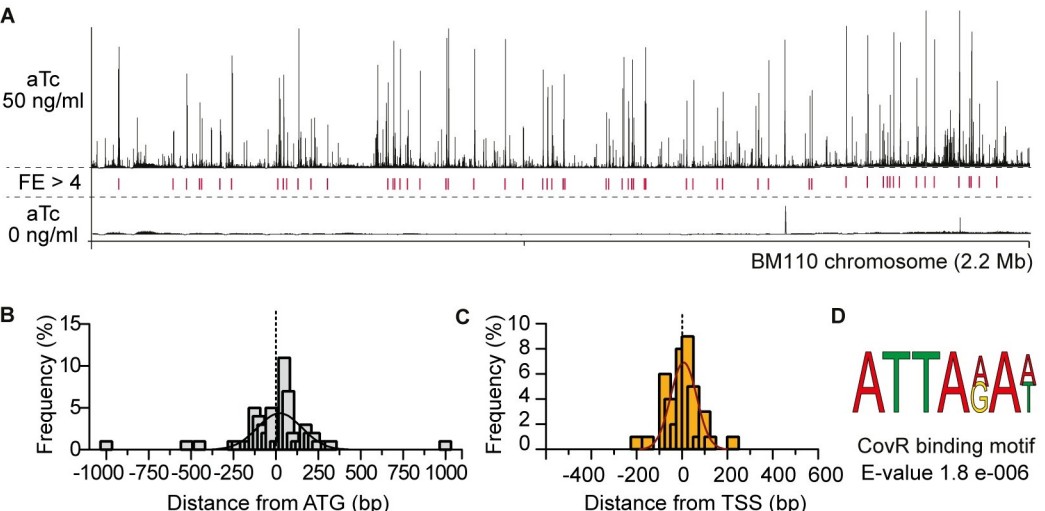

**Fig 2. Whole-genome CovR binding on the BM110 genome.** (A) ChIP-seq profile of CovR on the BM110 chromosome. Sequence reads were mapped on the chromosome after induction of the epitope-tagged FLAG-CovR with 50 (upper panel) or 0 ng/ml (bottom panel) anhydrotetracycline (aTc) in a Δ*covR* mutant. Peak height represents the mean coverage at each base pair of two independent ChIP-seq experiments. Loci with significant fold enrichment (FE > 4, IDR < 0.05) are indicated by red lines. (B) Distribution of the distance between each CovR binding peak and the nearest start codon. Distances were calculated from the summit of each CovR peak. The histogram represents the proportion of CovR binding sites (N = 62) in each sliding window of 25 bp, with an additional fitting curve. (C) Distribution of the distance between each CovR binding peak and the nearest transcriptional start site. Calculated as for (B). (D) Predicted CovR binding consensus sequence. Sequence enrichment in the 62 CovR binding loci (100 bp each) identified with the DREME software [86].

rich sequence is similar to the motif previously identified by conventional footprint experiments in *S. agalactiae* [21] and *S. pyogenes* [44], but is not sufficient to accurately predict CovR binding sites on the chromosome [41].

## CovR directly coordinates virulence genes expression

To characterize the direct CovR regulon, we combined the CovR binding loci identified by ChIP-seq with the transcriptional data. We chose strict criteria to define a conservative CovR direct regulon by only considering genes or operons differentially transcribed in the two *covR* mutants with a CovR binding located near the TSS (+/- 100 bp) and/or the first start codon (-200 to +100 bp). This direct CovR regulon includes 21 binding loci regulating the expression of 51 genes (Table 1 and S2B Table). All genes have an increased transcription in *covR* mutants, demonstrating the primary role of CovR as a transcriptional repressor. Notably, all but one of the highly repressed genes (N = 33/34 with Log2 FC > 3) are directly regulated by CovR (Fig 1 and S1F Table).

At the functional level, CovR directly represses cell-wall and secreted proteins involved in GBS pathogenesis, including the HvgA, FbsA, PbsP, and Pil-1 adhesins, the C5a peptidase ScpB and related serine proteases, and the secreted NucA endonuclease and FbsB adhesin (Table 1). In addition, CovR directly represses the *cyl* operon, as well as five secreted peptides or proteins. The remaining 15 proteins of the CovR direct regulon are membrane-localized or cytoplasmic (Table 1). These proteins likely contribute to the adaptation of GBS to the host by linking virulence gene expression with metabolic uptake (amino-acid ABC transporter), proteogenic stress (*e.g.* polyphosphate kinase [45,46]) and quorum-sensing (RgfAC two-component system and PepO endopeptidase [47–49]).

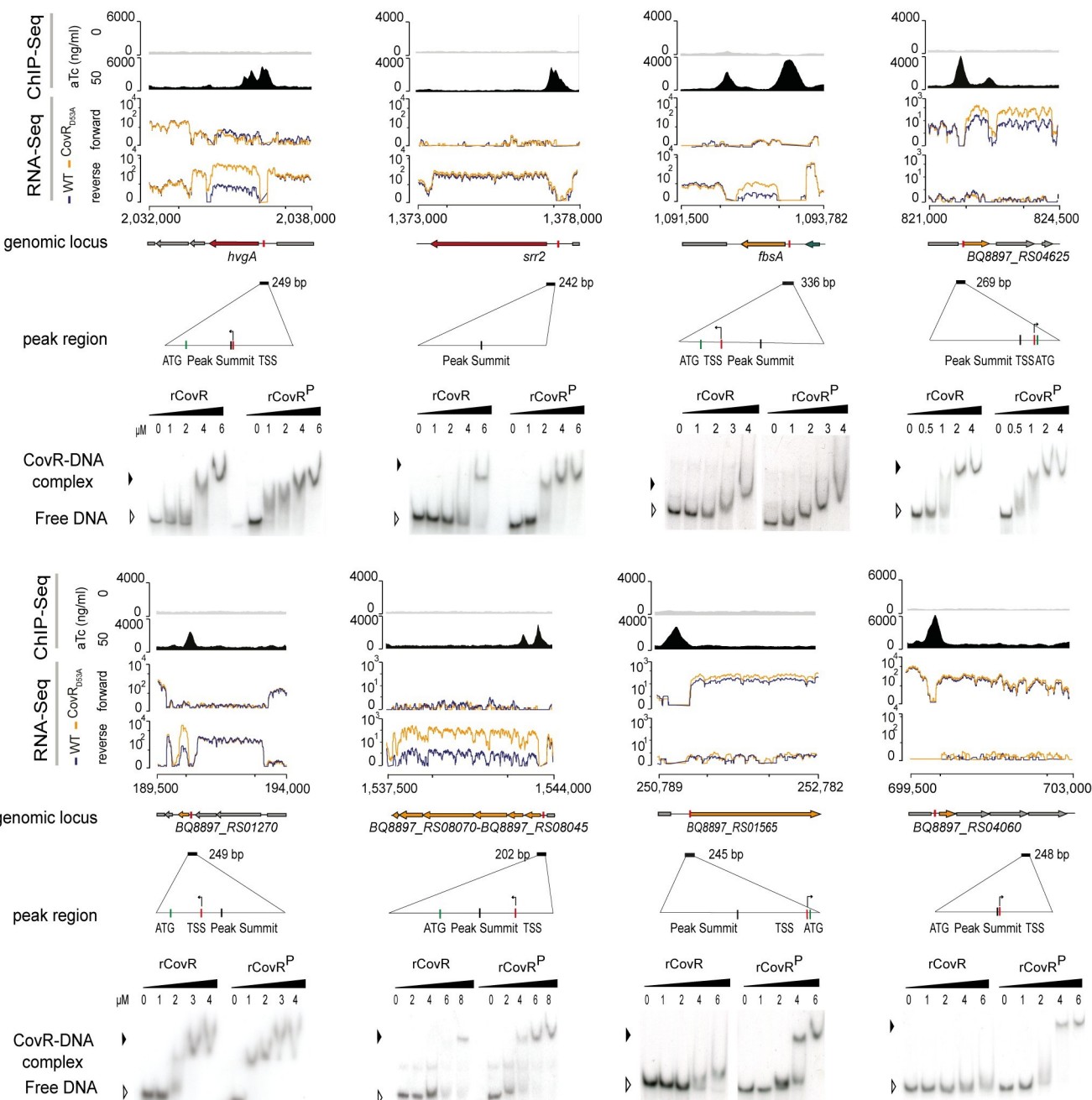

**Fig 3. Direct and complex CovR transcriptional regulation.** ChIP-seq and RNA-seq profiles showing CovR binding and the associated transcriptional signal for eight selected loci. From each locus is shown from top to bottom: *i)* the non-induced (anhydro-tetracycline (aTc) = 0: upper gray line) and induced (aTc = 50 ng/ml: bottom dark profile) ChIP-seq profiles with the normalized sequencing coverage scale indicated on the left axis; *ii)* the normalized strand-specific RNA-seq profiles of the WT (blue) and CovR$_{D53A}$ mutant (yellow) of the same genomic region with the chromosomal coordinates; *iii)* the schematic representation of ORFs in the locus, with the name of the regulated genes; *iv)* the position and size (202 to 336 bp) of sequences encompassing the ChIP-seq peaks, with a zoom in to highlight the position of the peak summits, the TSS, and the ATG; and *v)* the validation of CovR binding by EMSA with the recombinant purified rCovR. The formation of a rCovR-DNA complex is visualized by a delayed migration of the radiolabeled probe compared to the unbound probe (free DNA) and the affinity of rCovR to DNA is increased upon its phosphorylation by acetyl phosphate (denoted as rCovR$^P$). Note that for three loci (*srr2*, *BQ8897_RS01565*, *and BQ8897_RS04060*), CovR binding is not associated to a significant transcriptional change in the CovR$_{D53A}$ mutant.

**Table 1. The CovR direct regulon in BM110.**

| First gene in Transcriptional Units (TUs) | Number of genes in TUs | ChIP-seq (FE) | RNA-seq (mean Log2 FC) | Genes included in TUs | Main functions |
|---|---|---|---|---|---|
| | | | | | **LPxTG proteins** |
| BQ8897_RS10700 | 1 | 6,3 | 7,6 | *hvgA* | hypervirulent adhesin |
| BQ8897_RS05875 | 1 | 7,0 | 4,2 | *fbsA* | Adhesin |
| BQ8897_RS02735 | 1 | 4,2 | 1,5 | *pbsP* | Adhesin |
| BQ8897_RS03985 | 2+6 | 6,3 | 3,0 | *pil-1* operon | Adhesin and regulator (2 TU) |
| BQ8897_RS06775 | 1 | 5,5 | 1,7 | *scpB* | Serine protease (C5a peptidase) |
| BQ8897_RS02910 | 1 | 7,2 | 5,4 | *scpB2* | Serine protease (frameshifted) |
| BQ8897_RS04140 | 4 | 6,6 | 1,5 | *scpB4A* | Serine protease + Secreted small protein (134 aa) |
| | | | | | **Secreted** |
| BQ8897_RS04080 | 12 | 4,3 | 4,6 | *cyl* operon | ß-hemolysin/cytotoxin synthesis and export |
| BQ8897_RS04945 | 2 | 6,7 | 7,7 | *fbsB* | Adhesin |
| BQ8897_RS04215 | 1 | 4,4 | 3,7 | *nucA* | DNA/RNA non-specific endonuclease |
| BQ8897_RS02690 | 1 | 6,1 | 5,9 | | Predicted esterase/lipase |
| BQ8897_RS01270 | 1 | 4,2 | 7,5 | | Uncharacterized peptide (49 aa) |
| BQ8897_RS02625 | 1 | 6,3 | 5,0 | | Uncharacterized small protein (141 aa) |
| BQ8897_RS09870 | 1 | 6,4 | 6,5 | | Uncharacterized protein (586 aa—ATG miss-annotation in RefSeq) |
| | | | | | **Membrane and cytoplasmic** |
| BQ8897_RS10135 | 2 | 5,3 | 2,1 | *rgfAC* | Two-component system |
| BQ8897_RS09785 | 1 | 6,4 | 2,9 | *pepO* | Cytoplasmic endopeptidase (M13 family) |
| BQ8897_RS08070 | 6 | 5,7 | 6,7 | *pppK* | Uncharacterized—polyphosphate kinase |
| BQ8897_RS04650 | 3 | 8,1 | 2,0 | | ABC transporter (Amino acid transport) |
| BQ8897_RS05655 | 1 | 7,1 | 3,1 | | Membrane protein (Abx1-like) |
| BQ8897_RS04625 | 1 | 7,2 | 2,6 | (*fbsC\**) | Membrane protein (co-regulation with the frameshifted FbsC adhesin) |
| BQ8897_RS03040 | 1 | 4,4 | 3,1 | | Methyltransferase (frameshifted) |

The direct regulon might also include the gene encoding the FbsC adhesin (Table 1). However, since the FbsC adhesin is not functional due to conserved frameshift mutations in CC-17 strains [50], we did not investigate further its co-regulation with the *BQ8897_RS04625* gene (Fig 3). Additionally, we manually inspected the ChIP-seq profiles at the chromosomal *covR* locus. We did not observe any signal corresponding to CovR binding on its own promoter, arguing against a CovR feedback loop in BM110 [26].

## Indirect and secondary regulations by CovR

Our strict definition of the CovR direct regulon encompassed 21 out of the 62 chromosomal binding sites. The remaining binding sites were sorted into a larger regulon consisting of 3 groups (Tables 2 and S2B). The first group includes genes differentially transcribed in one of the two *covR* mutants only, while the second group is not associated with significant transcriptional changes in *covR* mutants (Table 2). We validated CovR binding by EMSA on the operon promoters of *srr2* and of two transporters (*BQ897_RS04060* and *BQ897_RS01565*). The binding of rCovR and phosphorylated rCovR to these promoters did not differ from the binding to promoters of the direct regulon (Fig 3). Next, we asked whether the functions encoded by the extended regulon differ from those of the direct regulon, but both encode for similar functions mainly associated with virulence and metabolites transport (Tables 2 and S2B). The most likely

**Table 2. The extended CovR signaling pathway in BM110.**

| Category | CovR binding (FE >4) | Transcription in *covR* mutants | Number of binding loci | Number of genes | Proposed mechanism | Main genes | Main functions |
|---|---|---|---|---|---|---|---|
| Extended regulon (group 1) | Yes | Up (mutant-specific) | 17 | 45 | CovR co-regulation | *srr2*, *sapA*, *fhuCDBG*, *nox*, *shtII*, *adcAII* | Transporters (6), Transcriptional regulator (2), LPxTG (2), secreted proteins (4), oxidative stress (3) |
| Extended regulon (group 2) | Yes | No | 12 | 23 | CovR co-regulation | *scpB4B*, *secA* | Transporters (5), LPxTG (1), secretion (3), antisens RNA (1) |
| Atypical (group 3) | Yes | Variable | 12 | 31 | Silencing | *csp*, *lpxB* | Prophages, Transposase, capsule |
| Indirect regulation | No | Up | 0 | 20 | Downstream regulation | *rib*, *lep* | Transport (3), Riboflavin synthase, Cell wall synthesis (1), secreted (1) |
| Positive regulation | No | Down | 1 | 3 | DNA Topology? | *cfb* | CAMP factor and DNA Topology modulation protein |

explanation for the discrepancies between CovR binding and transcriptional data is a complex regulation involving additional transcriptional activators, probably overlapping (group 1) or not (group 2) the CovR binding sites. The integration of CovR signalling into a wider network of regulators is also evident by the indirect regulation (*i.e.* significant transcriptional changes in the *covR* mutants not associated with CovR binding) of 20 out of the 79 genes in the CovR regulon (S1F Table).

The third group includes the remaining 12 binding loci localized inside ORFs, in 3' intergenic region, or associated with specific mobile elements (Tables 2 and S2B). Interestingly, these regions are often variable in the GBS population and several are important for pathogenesis, including the capsule operon (two CovR binding peaks in the middle and at the 3' end of the operon), the Tn*916* element containing the *tetM* resistance gene, and the *scpB-lmb* locus. Notably, CovR binding is detected in prophages with atypical clusters of genes differentially transcribed in the *covR* mutants (S2 Fig). Closer examination of these four loci showed significant CovR binding signals, either sharp (FE > 4) or diffused (1 < FE < 4), associated to a change in the transcriptional profiles of these mobile elements (S2 Fig). This suggests that non-phosphorylated CovR contribute to the silencing of recently acquired DNA regions and that new CovR binding sites might be selected to control the expression of advantageous genes. A non-canonical mechanism of regulation might also operate for the CAMP operon, the only highly positively ($Log_2$ FC < -5) CovR-regulated genes (Table 2). The promoter of the CAMP operon is the only one to display a significant ChIP-seq signal in the no-aTc control (Fig 2A), and was therefore excluded from the analysis. A second specific CovR binding signal is detected in the intergenic region and is associated to the divergently translated gene encoding a predicted nucleoid-associated protein [51], which may indicate a binding mechanism depending on DNA conformation rather than a consensus motif.

## CovR-regulated genes and promoters are under selective pressure

To identify genes directly regulated by CovR and subject to selective pressures, we took advantage of the previous reconstitution of the evolutionary history of the CC-17 lineage [3,4]. The hypervirulent lineage is a homogenous clonal complex adapted to the human host and the evolutionary pressure driving adaptation has been previously measured in the human-associated GBS population [4]. In total, 24 genes associated with CovR binding accumulated more mutations than expected under a neutral model of evolution, including 15 in CC-17 specifically

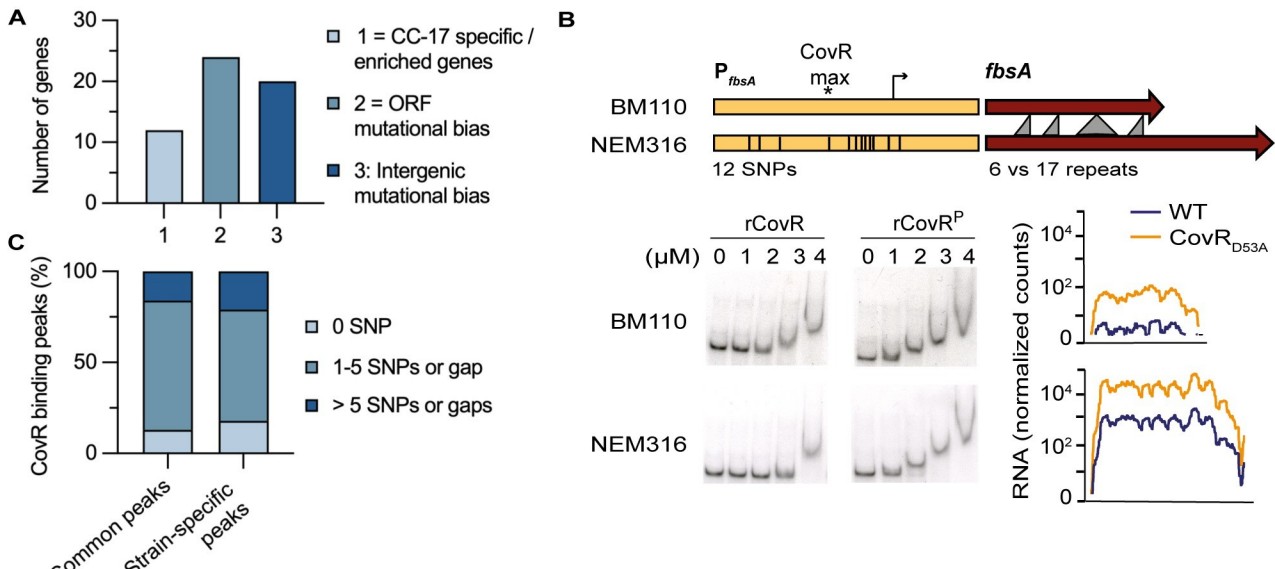

**Fig 4. Adaptation of CovR signaling in the hypervirulent lineage.** (A) Histogram reporting the number of CovR binding loci on the BM110 chromosome associated with CC-17 specific or enriched genes (category 1), with genes showing a mutational biases indicative of adaptive evolution (category 2), and in intergenic regions with mutational biases suggestive of CovR rewiring in CC-17 (category 3). The ORFs and intergenic mutational biases in the whole GBS population have been previously calculated to reconstitute the evolution of the CC-17 hypervirulent lineage [4] and the detailed list with gene IDs is in S4 Table. (B) CovR regulation of the FbsA adhesin encoding gene in BM110 (CC-17) and NEM316 (CC-23) strains. Schematic representation of the *fbsA* gene (dark red) and promoter (yellow) in the two strains showing the gene length difference leading to the translation of a protein with a different number of a repeated motif, and the 12 SNPs in the promoter region (336bp). Bottom: binding of non-phosphorylated and phosphorylated rCovR to the two promoters by EMSA and normalized RNA-seq data for the two WT strains and their corresponding CovR$_{D53A}$ mutants. Note that the two strains differ by their basal level of *fbsA* transcription and not by CovR binding or regulation. (C) Promoters mutations in direct CovR regulated genes. Cumulative percentage of CovR binding loci (250 bp centered on the maximum ChIP-seq signal) with 0 (black), up to five (grey) or more than five SNPs or gaps (white) between BM110 and NEM316 sequences. Common peaks = CovR binding loci detected in BM110 and NEM316; Strain-specific peaks = CovR binding loci detected in one strain only. Sequences alignment was performed by blastn.

(Fig 4A and S4 Table). Additionally, a signature of positive selection with an increased frequency of non-synonymous versus synonymous mutations (dN/dS > 1) was noticeable in *srr2*, *scpB*, and a gene of the capsule operon (S4 Table). A similar analysis on the intergenic regions identified 85 noncoding regions showing a significant mutational bias in the hypervirulent lineage [4]. Twenty of these intergenic regions were associated with a CovR binding locus, including 9 in the CovR direct regulon (Fig 4A and S4 Table), potentially resulting in different CovR binding at several loci and CovR rewiring on a global scale.

The CC-17 specific pan-genome is composed of 70 genes [4]. Among them, CovR binds on the promoters of the two operons encoding the HvgA and Srr2 hypervirulent adhesins as well as on the promoter of *scpB4B* encoding a non-functional serine protease (S4 Table). In addition to the frameshift in the *scpB4B* gene, several CovR-regulated genes also encoded for non-functional proteins due to frameshift mutations or SNPs generating internal stop codon. These mutations are localized in genes encoding proteins usually involved in GBS-host interaction such as the ScpB2 serine protease, the FbsC adhesin, a secreted endonuclease and two transporters (S1F and S2A Tables). This indicates that pseudogenization might be as important as the acquisition of new adhesins in reshaping the interaction of CC-17 with its human host. Overall, the CovR network appears to evolve rapidly with the gain or loss of genes, either by acting on promoters or on the gene sequences, probably as a consequence of the host selective pressure.

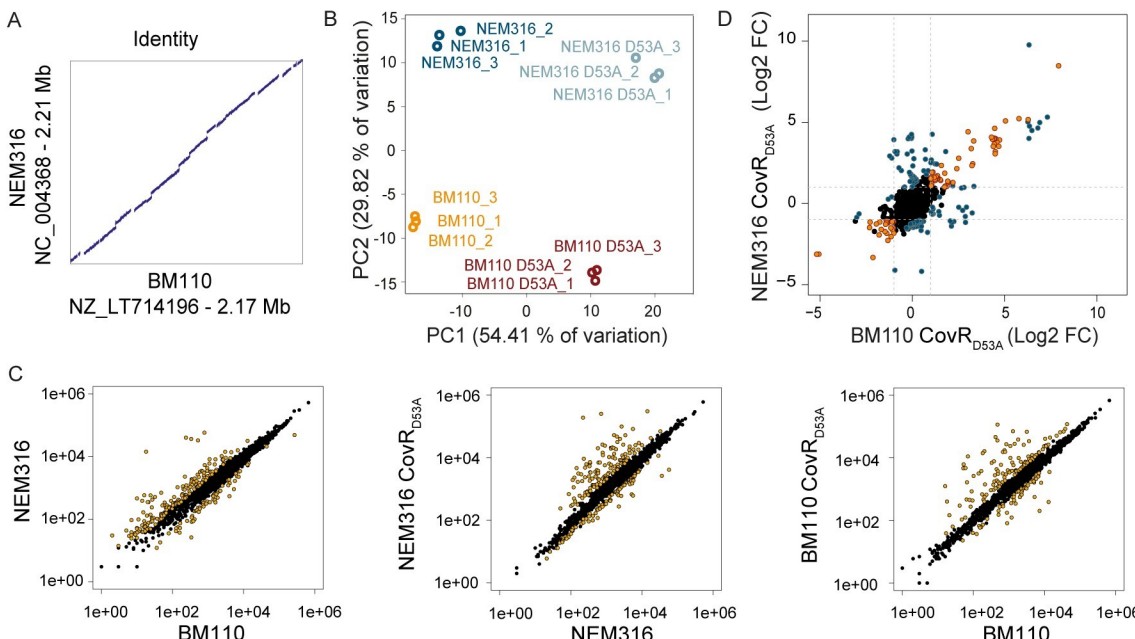

**Fig 5. Plasticity of the CovR regulon in BM110 and NEM316.** (A) Sequence identity between the NEM316 and BM110 chromosomes. The two chromosomes are co-linear with vertical and horizontal gaps corresponding to strain-specific sequences, usually due to prophages and ICE. (B) Principal component analysis of RNA-seq data for the BM110 and NEM316 WT strains and their corresponding CovR$_{D53A}$ mutants. Biological triplicates are resolved by PCA with variability sustained by CovR inactivation (PC1) and WT differences (PC2). (C) Scatter plots of the normalized read counts for each of the 1,716 homologous genes in the two WT strains, and between the WTs and their corresponding CovR$_{D53A}$ mutants. Yellow dots represent significant differential transcription (adjusted p-value < 0.005). (D) Pairwise comparison of the fold change in the two CovR$_{D53A}$ mutants. Genes with a similar and significant fold change upon CovR inactivation in the two backgrounds are symbolized with orange dots. Genes with a significant fold change in one mutant only or which show a significant different fold change in the two mutants are highlighted with blue dots.

## The plasticity of CovR signalling reshapes virulence gene expression

To quantify the specificities of CovR regulation in two strains, we did parallel RNA-seq experiments in BM110 and NEM316 (serotype III, CC-23) [52] backgrounds. The transcriptional profiles of CovR$_{D53A}$ mutants in BM110 and NEM316 are globally similar, with a large-scale transcriptome remodelling and a distinctive set of highly activated genes (S5A and S5B Table). Comparative analysis on the 1,716 orthologous genes revealed distinct transcriptomes for both the WT strains and for their corresponding CovR$_{D53A}$ mutants (Fig 5). Inactivation of CovR accounts for 54.4% of the variability (PC1) between samples, while 29.8% of the variability (PC2) is sustained by WT specificities (Fig 5B).

The difference between the WT strains implies 172 and 60 genes with a significant increased or decreased expression ($|Log_2 FC| > 1$; adjusted p-value < 0.005), respectively, in NEM316 compared to BM110 (Fig 5C and S6A Table). The highest difference was observed for the expression of the direct CovR regulated gene *fbsA*, with nearly 1,000-fold induction ($Log2 FC = 9.42$, adjusted p-value $< 10^{-80}$) in NEM316 compared to BM110. The *fbsA* promoter differs by 12 SNPs in the two strains, suggesting a case of CovR signalling evolution. However, *in vitro* binding of rCovR is similar on the two promoters and CovR repression is conserved in the two strains (Fig 4B). Therefore, the SNPs in the *fbsA* promoters do not have a direct effect on CovR binding and regulation *per se* but should be related to the gain (in NEM316) or loss (in BM110) of a binding site for an additional transcriptional activator.

The CovR$_{D53A}$ core regulon encompasses 100 genes differentially transcribed in the same direction (activation or repression; |Log$_2$ FC| > 1; adjusted p-value < 0.005) in the two strains (Fig 5D and S6B Table). For 17 of these genes, the fold change associated to CovR inactivation is significantly different in the two backgrounds, and 92 additional genes are differentially expressed only in one of the two CovR$_{D53A}$ mutants (Fig 5D and S6C Table), highlighting strain-specific CovR regulation. The plasticity of the CovR regulatory pathway is especially striking for the transcription of genes encoding cell-wall anchored proteins involved in host-pathogen interaction [11]. In total, we identified 27 LPxTG proteins encoding genes localized in the core or accessory BM110 genome (S7 Table). The transcription profiles of LPxTG encoding genes in the two CovR$_{D53A}$ backgrounds reveals a strain-specific remodelling of the bacterial surface (S4 Fig). The strain differences involve the conserved transcription of allelic variants (*bibA/hvgA*), the loss-of function mutations in highly CovR regulated genes (*scpB2* and *fbsC* in BM110, PI-1 pili operon in NEM316) and significant transcriptional difference (*pbsP*, *nudP cdnP*, *srr1/srr2 locus*) (Fig 4).

## Strain-specificities depends on the level of CovR activation

To compare CovR binding on a genomic scale, ChIP-seq experiments in the NEM316 and BM110 backgrounds were done in parallel with two levels of CovR induction (50 and 200 ng/ml aTc) (S8 Table). In total, we detected 31 common loci associated with CovR binding in the two strains, which delineate a minimal conserved binding regulon (Fig 6A and S9A Table). In addition, reproducible CovR binding is observed specifically at 29 and 6 chromosomal loci in BM110 and NEM316, respectively (S9A Table). As expected, strain-specific CovR binding occurs at the level of specific genes, such as *srr2* in BM110, and in genes localized into non-shared mobile elements. However, strain-specific CovR binding also occurs at the level of 28 loci present in the two strains. While promoters of two transporters have large deletions (68 and 73 bp in BM110 and NEM316, respectively) or SNPs, which might explain CovR binding differences, 5 binding regions are identical in the two strains on 250 bp surrounding CovR binding (Fig 4C and S9B Table).

The ChIP-seq profiles suggested a global difference in the capacity of CovR to bind chromosomal DNA between strains. This difference is most striking by comparing individual ChIP-seq experiments (S9A Table). Analysis of ChIP-seq done with a low level of CovR induction (50 ng/ml aTc) revealed 25 significant peaks in NEM316 compared to 62 in BM110 (FE > 4, IDR < 0.05), with 15 common CovR binding regions between the two strains (Figs 6A and S5). By applying less stringent criteria (FE > 1, IDR < 0.05), a total of 292 and 324 significant CovR binding signals were detected in BM110 and NEM316, respectively (S10 Table), indicative of weak CovR interactions globally distributed along the chromosomes and specific CovR enrichment at regulated promoters, especially in BM110. In agreement, closer examination of the ChIP-seq profiles done with 50 ng/ml aTc revealed a lower signal to noise ratio in NEM316 compared to BM110 (Fig 6A). In contrast, induction of CovR with 200 ng/ml aTc increases the quantity of CovR (Fig 6B) and enhances the overall binding signal in NEM316 while it tends to increase the background signal in BM110 (Fig 6A). In this condition, 59 and 70 CovR significant peaks (FE > 4, IDR < 0.05) were detected in NEM316 and BM110, respectively (S5 Fig and S9A Table), mitigating the difference between strains.

To explain this difference, we hypothesized that CovR might be phosphorylated at different levels in the two strains grown under standard conditions. Indeed, analysis of CovR phosphorylation by Phos-Tag showed that up to 50% of CovR is phosphorylated in BM110 compared to less than 10% in NEM316 (Fig 6C and 6D). When expressing a FLAG-CovR$_{D53A}$ variant, only residual CovR phosphorylation is detected in the two backgrounds (Fig 6D), probably due to

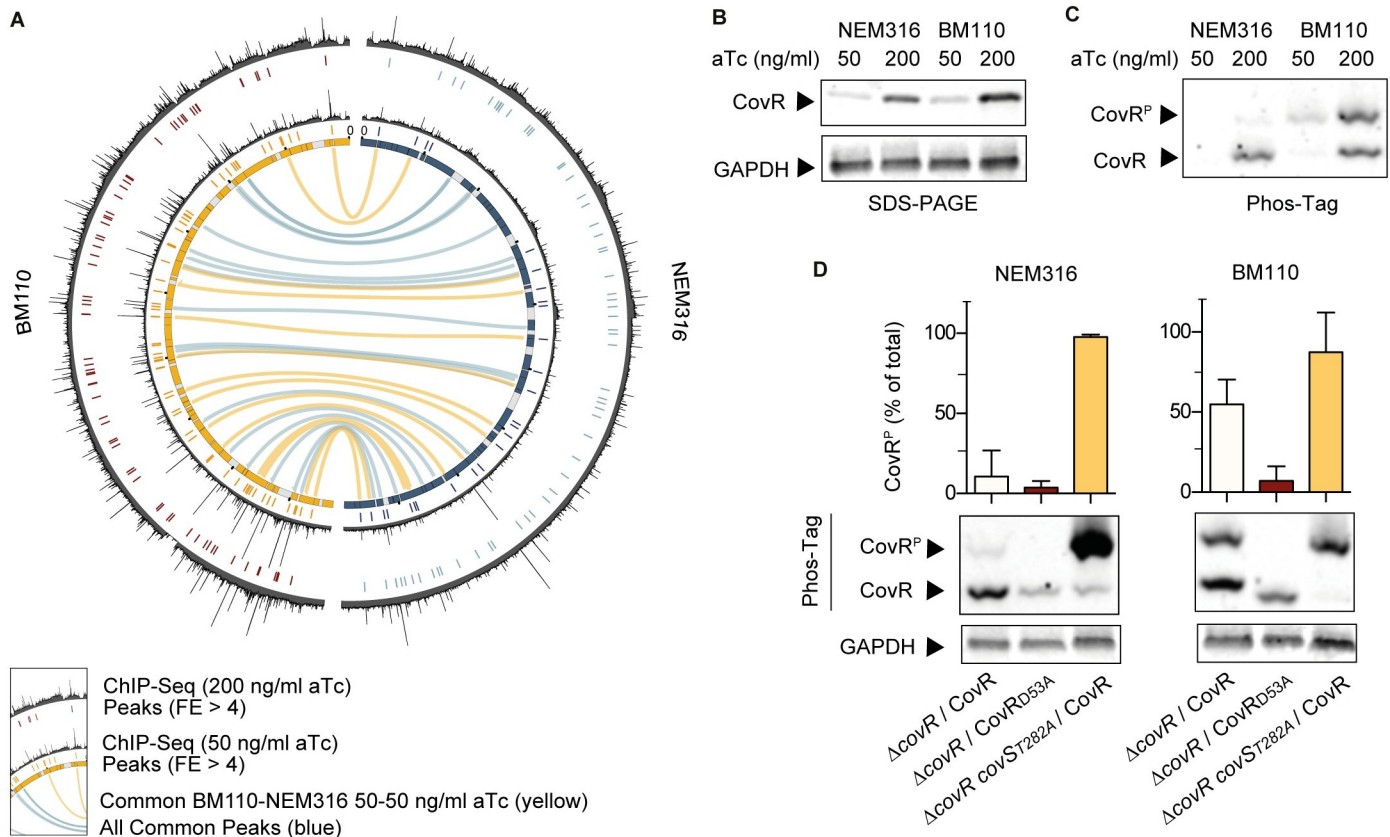

**Fig 6. Global effect of CovR phosphorylation on chromosomal binding.** (A) Shared CovR binding sites identified on the BM110 and NEM316 chromosomes. The inner circle is a symmetric representation of the BM110 (left, yellow) and NEM316 (right, blue) chromosomes with strain-specific sequences ($< 90\%$ homology) in grey. ChIP-seq profiles for each strain are shown after induction of FLAG-CovR with 50 (inner profile) or 200 (outer profile) ng/ml anhydro-tetracycline (aTc). Significant CovR peaks (FE $> 4$, IDR $< 0.05$) are symbolized by colored traits below each ChIP-seq profile. Conserved chromosomal binding loci are represented by the inner connecting lines (in yellow: between ChIP-seq done with 50 ng/ml aTc; in blue: all ChIP-seq experiments). (B) Dose-dependent induction of FLAG-CovR by aTc in BM110 and NEM316 strains. Western blot analysis with anti-FLAG (upper panel) and anti-GAPDH (bottom panel) antibodies after SDS-PAGE electrophoresis of total protein extracts prepared from mid-exponential growing cultures in THY. (C) Level of CovR activation by phosphorylation is strain-specific. The same extracts as in (B) were analyzed with anti-FLAG antibodies after Phos-Tag electrophoresis. A delay migration denotes a phosphorylated form (CovR$^P$). (D) Quantification of CovS-dependent CovR phosphorylation in NEM316 and BM110. The proportion of phosphorylated epitope-tagged CovR variants were analyzed by Phos-tag analysis after induction with 200 ng/ml aTc. The FLAG-CovR variant was used in the $\Delta covR$ (first rows) and $\Delta covR\ covS_{T282A}$ (third rows) mutants, the T282A substitution abolishing the phosphatase activity of CovS. A FLAG-CovR$_{D53A}$ variant (second rows) unable to be phosphorylated by CovS was used in the $\Delta covR$ mutants. Quantification are means and standard deviation of 5 biological replicates, and a representative Phos-Tag gel with its GAPDH loading control is presented for each background.

the activity of the serine threonine kinase Stk1 on the CovR $T_{65}$ residue [24]. Conversely, expression of FLAG-CovR in a CovS$_{T282A}$ mutant, in which the phosphatase activity of the histidine kinase CovS is specifically abolished [27,28], increased CovR phosphorylation at nearly 100% in the two backgrounds (Fig 6D). These results showed that CovS is functional in the two strains but that the basal level of CovR activation is different. This difference leads to strain-specific CovR regulation associated with a global effect on CovR binding at the genomic scale.

## Discussion

In this study, we demonstrated that CovR is the direct coordinator of GBS-host interactions and that the individual CovR-regulated genes and the whole network are subject to the host selective pressure. This inherent plasticity of the CovR signalling pathway allows to generate

diversity in the bacterial population, ultimately leading to the selection of host-adapted strains associated with specific pathologies. Previously, mutations in CovR or in the regulators of CovR activity have been identified as a major determinant of the virulence heterogeneity of the closely related Group A *Streptococcus* (GAS or *S. pyogenes*) pathogen [32,53]. Especially, loss-of-function mutations in *covR*, *covS*, or the associated *rocA* regulator are associated with the unusual severity of invasive infections by the M3 serotype [54–56] and to within host evolution of GAS hyper-invasive isolates [20,31,57,58]. Similarly, hyper-invasive *covR/S* null mutants have been occasionally identified in GBS, especially in strains causing *in utero* infections [14]. However, *covR/S* loss-of-function mutations have a cost and decrease host-to-host transmission of the hyper-invasive isolates, hampering their dissemination in the community [31,58]. Beyond mutations in the CovR/S components themselves, the analysis of the CovR direct regulon demonstrates that the whole signalling pathway is highly plastic and sustains the evolution of successful clonal complex.

In the case of the human-adapted CC-17 clones, the hypervirulence against neonates has been linked to the expression of the HvgA and Srr2 specific adhesins [6,7,9]. Here, we show that the both adhesins are directly regulated by CovR. This was expected for the *hvgA* gene, which is an allelic variant of the *bibA* gene present in non-CC17 clones transcribed from a conserved promoter [7,21]. In contrast to the *hvgA*/*bibA* variants, the *srr1*/*srr2* operons are distantly related, are localized at different genomic location, and are mutually exclusive [9,59]. Our results show that Srr2, but not Srr1, is embedded in the CovR network. Interestingly, CovR binds to the *srr2* promoter but only the absence of CovR leads to *srr2* up-regulation while the $CovR_{D53A}$ variant is sufficient to repress *srr2* transcription. The most likely hypothesis is that *srr2* transcription requires an activator that outcompetes the binding of the non-phosphorylated CovR form, which has a lower affinity for DNA than the phosphorylated form. The regulation of the related Srr1 serine-rich repeat adhesin in non CC-17 clones depends on the Rga transcriptional factor [9,59], but no Rga homolog is present near the *srr2* locus or in the CC-17 genomes suggesting the involvement of an yet-to-be identified activator and a different regulatory logic. This specific regulation should have evolved to preferentially express Srr2 in the gut to give an advantage during the initial colonization steps rather than during the invasive phase [8,60].

A co-dependence between CovR repression and specialized activators is evident at several loci in addition to *srr2*. For instance, the PbsP adhesin encoding gene is directly activated by the SaeRS two-component system to promote vaginal colonization [61], while *pbsP* transcription is also necessary at later stages of the infection process [62–64]. A second example is the transcription of the gene encoding the FbsA adhesin which depends on 3 regulators: RogB, RovS and Rgg [33,65–67]. Among them, the activator RogB is present in NEM316 but absent in BM110, suggesting a genetic basis for the difference in the basal level of *fbsA* transcription between the two strains. Lastly, CovR directly repress transcriptional activators, such as the Ape1 activator of the PI-1 pili locus and the RgfAC two-component system. These two activators are necessary for pathogenesis but, as other CovR regulated genes, are often mutated in the GBS population [4,23,47,48], introducing strain-variability in the CovR network.

The CovR/S two-component system belongs to the PhoP/Q family and, as expected for a canonical two-component system [68], CovR phosphorylation increases its affinity for DNA. The phosphorylated PhoP-like response regulators bind to complex promoters with binding sites that can vary in number, sequence, location and orientation [69,70]. This variability ensures a dynamic expression of the regulated genes, with promoters having the highest affinity for the regulator being the first to be activated or repressed [69]. The difference in CovR phosphorylation state between the two GBS strains offers a glimpse into this temporal hierarchy of regulated genes. The CovR binding regulon in BM110 is likely close to be exhaustive

while in NEM316 only promoters with the highest affinities for a phosphorylated CovR might have been identified [69]. An interesting instance is the *cyl* operon encoding the ß-h/c toxin which is directly regulated by CovR [21,22,71]. *In vivo* binding of CovR is highly significant in the BM110 strain, but is below the threshold (FE > 4) in the NEM316 strain. This correlates with the lower haemolytic activity and the higher level of CovR phosphorylation in BM110 compared to NEM316. The paradox of a more efficient repression of virulence genes in the hypervirulent clone might in fact reflect the clinical characteristic of CC-17 strains which are associated with a delayed colonization of the neonatal gut rather than an increase pathogenicity caused by toxin expression [8,60,72].

The difference in the basal level of CovR phosphorylation has likely an effect on the dynamics of the response to external stimuli, for instance in the acidic phagolysosome [73,74]. However, the signal(s) activating or inhibiting CovS is not yet identified in GBS, unlike in GAS in which the CovS homologue senses and responds to magnesium and to the human LL-37 antimicrobial peptide [28,40,41]. In addition to CovS, CovR activation in GBS is mediated by two additional proteins, the serine-threonine kinase Stk1 [24] and the CovS-interacting protein Abx1 [27]. While Stk1 and Abx1 are identical in the NEM316 and BM110 strains, CovS differs by one residue between the two strains. We cannot exclude at this stage that the CovS polymorphism, a valine to alanine substitution at position 112 localized into the extracellular loop, is the cause of the differential CovR phosphorylation in the two strains. An alternative hypothesis will be that CovR phosphorylation depends on an additional, but yet uncharacterized, regulator. Deciphering the mechanism of CovR activation and its variation in the GBS population is therefore essential to accurately compare the evolution of the CovR regulatory pathway.

In addition to the transcriptional repression of specific genes, our CovR transcriptome analysis suggests a non-canonical regulation of mobile genetic elements. Previously, the CovR homolog in *Streptococcus mutans* has been proposed to silence a pathogenicity island [75]. Notably, the *S. mutans* CovR is an orphan regulator which has lost its cognate histidine kinase CovS and, consequently, the CovR function is independent of its phosphorylation [75,76]. Instead, CovR acts as a competitor with a nucleoid-associated protein (NAP) to repress transcription [75,76]. This non-canonical regulation appears conserved in GBS with clusters of prophages genes showing an opposite regulation in the Δ*covR* and CovR$_{D53A}$ mutants. The regulation of mobile elements by NAPs or by the recruitment of an ancestral regulatory network, such as PhoP/Q or OmpR/EnvZ two-component systems, have been extensively described in gram-negative bacteria [29,77,78], but not in low-GC% gram-positive bacteria. Our results suggest that CovR binding and regulation might evolved rapidly to include recently acquired genes in the virulence network, but the mechanism as a silencer or anti-silencer requires further investigations [79].

## Conclusions

Virulence and pathogeny of streptococci widely vary between strains of the same species. A significant part of this phenotypic diversity is due to genomic differences, especially the acquisition of new virulence genes and allelic variability. In this study, we show that the master regulator of virulence CovR directly regulates a large array of virulence associated genes and that the whole regulatory network is subject to selective pressure generating diversity in the streptococcal population. Increasing evidences suggest that regulatory evolution is a driving force in the emergence of clones associated with specific pathologies [30–32], either by transcriptional rewiring to control specific genes necessary in a new environment [30] or by mutations in global regulators to reshape the entire network [31]. Our analysis indicates that the CovR regulatory network varies both locally at multiple loci (gene acquisition, pseudogenization,

promoter mutation, co-activation) and globally to impact the cellular response (CovR activation, atypical mechanism of regulation at specific loci) in the GBS population. By directly coordinating host-pathogen interactions, this plasticity of the signalling pathway sustains the adaptation of GBS to new environments and allows the emergence of clones associated with specific pathologies.

## Materials and methods

### Strains and growth conditions

BM110 (Serotype III, CC-17) and NEM316 (Serotype III, CC-23) strains are representative of two of the five main GBS clades associated with human infections [3,52]. All strains and plasmids used in this study are detailed in the S11 Table, and standard growth conditions are defined as cultures in Todd-Hewitt Yeast (THY) buffered with 50 mM Hepes (pH 7.4) incubated at 37˚C in static condition. Columbia agar supplemented with 10% horse blood and Granada medium (BioMerieux) were used for propagation and for visualisation of ß-hemolytic activity and pigmentation, respectively. Erythromycin and kanamycin (Sigma-Aldrich) are used for plasmid selection and maintenance at 10 and 500 μg/ml, respectively, while anhydrotetracycline (Sigma-Aldrich) is used for conditional expression. For *Escherichia coli*, LB medium were used with ticarcillin (100 μg/ml), chloramphenicol (30 μg/ml), erythromycin (150 μg/ml), or kanamycin (25 μg/ml) when appropriated.

### Plasmids and strains construction

Oligonucleotides and plasmids construction are detailed in the S11 Table. Briefly, for the epitope-tagged expression vector, a synthetic DNA containing a translational initiation site, a start codon, the 3xFLAG epitope, a flexible linker, three stop codons in the three reading phases and a transcriptional terminator was synthesized and cloned into a pEX vector (MWG Genomics). An inverse PCR on the pEX vector and a PCR on genomic DNA, followed by Gibson assembly, were used for in-frame cloning of the *covR* sequence between the linker and the stop codons. The *covR*-synthetic DNA fragment was excised from the pEX vector by BamHI digestion and cloned into the aTc inducible expression vector pTCV_P$_{tetO}$ [50].

The construction of the Δ*covR* mutants were previously reported in NEM316 [27] and BM110 [64], as well the construction of the NEM316 CovR$_{D53A}$, and CovS$_{T282A}$ mutants obtained by chromosomal substitutions [27]. The BM110 point mutants were constructed as described in NEM316 with the pG1 shuttle thermosensitive plasmid [27]. For chromosomic complementation of the Δ*covR* and CovR$_{D53A}$ mutants, the WT *covR* allele with adjacent sequences (2 x 500 bp) was amplified and cloned by Gibson assembly into the pG1 vector. After GBS transformation of the pG-covR WT$_{repair}$ vector, chromosomal integration at the *covR* locus, de-recombination and loss of the vector, clones with a WT *covR* phenotype on Blood agar were selected and confirmed by PCR on genomic DNA. For mutations swapping, a pG-covR_D53A$_{swapped}$ vector was similarly constructed and used in the Δ*covR* mutant, and a pG-Δ*covR* vector was used to transform the CovR$_{D53A}$ mutant.

Genomic DNA of the parental strains and of *covR* mutants were purified (Qiagen DNeasy Blood and Tissue kits) and sequenced on an Illumina Miniseq following manufacturer instructions. Reads were mapped against the respective NCBI reference genomes and SNPs analysed with Genious prime (2019.2.3—Biomatters Ltd). Results of genome sequencing are summarized in S12 Table.

## RNA- and dRNA-sequencing

RNAs purification for RNA-seq were done from three independent cultures, with replica done in different days, in 10 ml of THY, 50 mM Hepes pH 7.4. RNA stabilization reagents (RNA-protect, Qiagen) were added at mid-exponential phase ($OD_{600}$ 0,5–0.6) for 5 min at ambient temperature. Cells were harvested at 4°C, washed with 1 ml cold PBS, and the bacterial pellets stored at minus 80°C. Cells were mechanically lysed and total RNA extracted following manufacturer instructions (FastPreps and FastRNA ProBlue, MP Biomedicals). Residual DNA were digested (TURBO DNase, Ambion) and samples qualities were validated (Agilent Bioanalyzer 2100, Qubit 3.0, Life Technologies) before rRNA depletion, libraries construction and sequencing (Ribozero rRNA, TruSeq Stranded mRNA, Hiseq2500, Illumina). RNA purification for dRNA-seq to determine TSS positions in the BM110 strain were done, processed and analysed exactly as described in the NEM316 strain [43]. The specificities of the dRNA-seq protocol are a Tobacco Acid Pyrophosphatase (TAP) treatment of RNAs and the use of a specific 5′ adapter to differentiated primary transcripts and processed RNAs. For qPCR analysis, RNAs were prepared as for RNA-seq. Standard reverse transcription and quantitative PCR were done as described (Biorad) [27].

For RNA-seq, single-end strand-specific 65 bp reads were cleaned (cutadapt version 1.11) and only sequences at least 25 nt in length were considered for further analysis. Alignment on the corresponding reference genomes (Bowtie v1.2.2 with BM110 RefSeq NZ_LT714196 and NEM136 RefSeq NC_004368) [80] and gene counts data (featureCounts, v1.4.6-p3, Subreads package; parameters: -t gene -g Name -s 2) were analysed with R (v3.6.1) and the Bioconductor package DESeq2 (v1.26.0) [81]. Normalization and dispersion were estimated and statistical tests for differential expression were performed applying the independent filtering algorithm. A generalized linear model including the replicate effect as blocking factor was set in order to test for the differential expression between the mutant and the WT strains. For each comparison, raw p-values were adjusted for multiple testing according to the Benjamini and Hochberg procedure [82] and genes with an adjusted p-value lower than 0.005 were considered differentially expressed. The coverage profiles were obtained for each strand using bedtools (v2.25.0), normalized using the DESeq2 size factors and then averaged across the biological replicates.

## Chromatin immunoprecipitation and sequencing

Cultures of the BM110 and NEM316 Δ*covR* mutants containing the pTCV-P$_{tetO}$-FLAG-*covR* or the pTCV-P$_{tetO}$-*covR* vector were done in parallel with independent duplicates for each condition. Overnight cultures in THY, Hepes 50 mM, kanamycin 500 μg/ml, were inoculated (1/50) in 100 ml of fresh media supplemented with the indicated concentration of aTc, and incubated until mid-exponential growth phase ($OD_{600}$ 0,5–0,6). Crosslinking were done by the addition of 1% formaldehyde for 20 min at room temperature under agitation, followed by quenching with 0,5 M glycine for 15 min. Bacteria were harvested, washed two times in ice-cold Tris-Buffered saline (20 mM Tris/HCl pH 7.5, 150 mM NaCl) and resuspended in 1 ml Tris-Buffered saline supplemented with protease inhibitor (cOmplete Protease Inhibitor, Roche). Bacterial lysis was done by enzymatic cell wall degradation (30 min at 37°C with 10K/ml mutanolysine, Sigma) followed by mechanical disruption (FastPrep-24, MP biotechnological) at 4°C with 0,1 mm glass beads (Scientific Industries, Inc). After centrifugation (4°C, 5 min), 500 μl of supernatants were diluted with 500 μl of cold immunoprecipitation buffer (50mM HEPES-KOH, pH 7.5, 150mM NaCl, 1mM EDTA, 1% Triton X-100, 1mM fresh PMSF) and the chromatin was fragmented by sonication (Covaris S220) for 20 min in milliTUBE (1ml with AFA Fiber). Aliquots of 100 μl were collected to measure DNA

fragmentation and to confirm CovR expression by agarose gel electrophoresis and Western blot with anti-FLAG antibodies, respectively.

Chromatin immunoprecipitation were done with 40 μl of Anti-FLAG M2 magnetic beads (Millipore) for 2h at 4°C under constant agitation. Beads were successively washed with four buffers (twice in immunoprecipitation buffer, twice in 50mM HEPES-KOH pH 7.5, 500mM NaCl, 1mM EDTA, 1% Triton X-100, 1mM fresh PMSF, once in 10mM Tris-HCl pH 7.5, 250mM LiCl, 1mM EDTA, 0.5% NP-40, 1mM fresh PMSF, and once in TE buffer 10mM Tris pH 7.5, 1mM EDTA). CovR elution were done in 50mM Tris-HCl pH 7.5, 1mM EDTA, 1% SDS pH 8.0 and confirmed with aliquots analysed by Western blot. Samples were treated with RNAse (Sigma) for 30 min at 37°C and reverse crosslinking was carried out by an overnight incubation at 65°C with 0,1 mg/ml proteinase K (Eurobio). Magnetic beads were discarded and DNA purified (QIAEX II, Qiagen) and quantified (Qubit 3.0, Invitrogen). DNA libraries preparation with 16 cycles of PCR amplification and sequencing were done following manufacturer instructions (TruSeq ChIP-Library kit, NextSeq 500/550, Illumina).

Quality controls, trimming and genome mapping of single end sequencing reads (75-bp) were first proceeded similarly to the RNA-seq reads. A step was applied to filter duplicated reads (Picard-tools v2.8.1, Samtools, v1.6) and a strands cross-correlation metrics step (phantompeakqualtools, R, v3.0.1) was applied for quality metrics and to evaluate fragment length before peak calling (Macs v2.1) [83,84]. The corresponding no-tag samples were used as controls and only peaks with a p-value inferior to 0.01 were considered. Reproducible peaks between independent replicates were identified with an expected rate of irreproducibility discovery threshold of 0.05 (IDR, v2.0.2) [85]. Functional assignation of peak summits to ATG and TSS were done (BEDtools v2.25.0) before manual validation on normalized read coverage generated with custom bash scripts and visualized with Integrated Genome Viewer (v2.3.25) and Geneious (Biomatters Ltd, v2019.2.3). DREME [86] was used to find enriched motifs using 100 bp of sequences centred on peak summits.

## Recombinant rCovR purification

For recombinant rCovR purification, the *covR* sequence was cloned into the pET-24a expression vector (Life Technologies) and transformed into BLI-5 *E. coli* strain for expression. Recombinant rCovR with a C-terminal histidine tag was purified from 800 ml of culture grown overnight at 20°C after induction with IPTG (0.1 mM) added during exponential growth ($OD_{600}$ = 0.8). Bacterial cells were collected by centrifugation, frozen at -20°C, resuspended in 40 ml of 50 mM Na2HPO4/NaH2PO4, 300 mM NaCl, pH 7.0, and lysed through two passages on a cell disruptor. After centrifugation, supernatants were filtered (0,22 μM Steriflip, Merck) and incubated for 20 min under constant rotation with 3,5 ml of pre-washed Ni-NTA superflow beads (Qiagen). Beads were collected by centrifugation, washed two times for 10 min with 20 ml of fresh buffer, and elution of rCovR from beads were done on the top of a size exclusion column (Biorad) with 50 mM Na2HPO4/NaH2PO4, 300 mM NaCl, 150 mM imidazole, pH 7.0. Fractions containing rCovR were pooled and desalted (PD-10 columns, GE Healthcare) with 50 mM Tris-HCl, 500 mM NaCl, pH 8. Aliquots of rCovR were conserved at -20°C in buffer supplemented with 30% glycerol, pH 8.

## CovR phosphorylation and *in vitro* binding

For *in vitro* rCovR phosphorylation, up to 5 μg of rCovR was incubated for 60 min at 37°C with 20 mM MgCl2 and 35 mM lithium salt acetyl-phosphate (Sigma), and phosphorylation was confirmed by Western analyses with anti-His tag antibodies after electrophoresis in 12.5% Phos-Tag SDS polyacrylamide gels (SuperSep Phos-Tag, Wako Pure Chemical Industries Ltd)

for 2 hours (100V, 30 mA) in Tris-glycine buffer on ice. The negative control is the heated (100˚C, 1 min) sample, removing the phospho-labile phosphoryl group. *In vivo* CovR phosphorylation were similarly analysed with rabbit anti-FLAG antibodies (1:1000) after electrophoresis of 20 μg of total GBS protein. Mouse anti-GAPDH antibodies were used as loading controls. Fluorescent secondary antibodies (goat anti-rabbit and anti-mouse IRDye 800 CW, Licor Biosciences) were used and signals revelation and quantification were done with Odyssey Imaging system (Licor Biosciences) on at least five independent protein extracts.

Electrophoretic mobility shift assay (EMSA) were done with PCR probes produced with a forward primer previously radiolabelled with [γ-32P]-dATP by the T4 polynucleotide kinase (New England Biolabs). Protein-DNA interaction was performed with variable concentrations of rCovR, radiolabelled probe, 0.1 μg/μl of Poly(dI-dC) (Pharmacia), and 0.02 μg/μl BSA in binding buffer (25 mM Na2HPO4/NaH2PO4 pH 8, 50 mM NaCl, 2 mM MgCl2, 1 mM DTT, 10% glycerol) for 30 min at room temperature. Samples were separated onto a 5% TBE-polyacrylamide gel for 1 h 30 min and revealed by autoradiography. For each probe, EMSA were done with the same aliquot of rCovR without and with extemporaneously phosphorylation. DNase I protection assays (footprinting) were done in similar condition and as previously described [21].

## Supporting information

**S1 Fig. Complementation of the *covR* mutant phenotypes by the inducible epitope-tagged FLAG-CovR variant.** (A) Pigmentation and ß-hemolytic phenotypes of *covR* mutants. Dilutions of overnight cultures ($10^{-5}$ and $10^{-6}$) of the BM110 (CC-17) and NEM316 (CC-23) wild-type strains and of their corresponding Δ*covR* deletion and $CovR_{D53A}$ substitution mutants were spotted on rich media (THY) and Columbia agar supplemented with horse blood (Blood). Photos were taken after 24–36 h of growth at 37˚C. (B) Schematic representation of the epitope-tagged FLAG-CovR system. The epitope-tag (3xFLAG) is linked in frame to the 5' end of the *covR* sequence by a flexible linker (Gly-Ala x 3). Transcription is dependent upon the binding of anhydro-tetracycline (aTc) to the trans-activator TetR, leading to the binding of TetR to the $P_{tetO}$ promoter containing tet operators sequences upstream the transcriptional start site. The *tetR* gene is cloned in the same pTCV-$P_{tetO}$ vector (not represented) as the $P_{tetO}$-FLAG-*covR* cassette. (C) Repression of negatively CovR-regulated genes by the epitope-tagged FLAG-CovR variant. Induction of FLAG-CovR in the BM110 Δ*covR* / pTCV-$P_{tetO}$-FLAG-*covR* strain was done with increasing concentration of aTc (0, 25, 50, 100, 250, 500 ng/ml) and RNA were prepared from mid-exponentially growing cultures. RT-qPCR were done on selected genes up-regulated in the parental Δ*covR* mutant and on the *covS* gene as a control. Results are reported as the fold change difference (log2 FC) between the induced and the non-induced (aTc = 0) condition, with *gyrA* as the reference gene. Mean and standard deviation are calculated from three biological replicates. (D) Phenotypic complementation of the pigmentation and ß-hemolytic phenotypes of the BM110 Δ*covR* mutant. The BM110 Δ*covR* / pTCV-$P_{tetO}$-FLAG-*covR* strain was spotted as in (A) on THY and Columbia Blood Agar without (-aTc) or with 500 ng/ml aTc supplementation (+aTc). (E) Dose-dependent induction of FLAG-CovR by aTc. Western analysis of total protein extracts with anti-FLAG and anti-GAPDH antibodies after cultures of the BM110 Δ*covR* mutant containing the pTCV-$P_{tetO}$-FLAG-*covR* expression vector with increasing concentration of aTc. (F) Comparative analysis of transcriptional repression by *covR* (upper panel) and FLAG-*covR* (bottom panel) cloned into the same pTCV-$P_{tetO}$ inducible vector into the BM110 Δ*covR* mutant. RT-qPCR were done starting with RNAs prepared from uninduced (aTc 0 ng/ml) and induced (aTC 50 and 200 ng/ml) cultures, with two biological replicates and technical triplicate. (G) Comparative

analysis of CovR and FLAG-CovR expression with anti-CovR antibodies. Western were done with 20 μg of total protein extracts of the BM110 WT, of the Δ*covR* mutant, and of the Δ*covR* / pTCV-P$_{tetO}$-FLAG-*covR* mutant grown with increasing concentration of aTc (0–200 ng/ml). Membranes were hybridized with anti-CovR antibodies (GenScript) and revealed with fluorescent secondary antibodies (upper panel). Loading controls are given by the non-specific hybridization signal (upper panel) and by Ponceau S coloration (bottom panel).
(TIF)

**S2 Fig. Atypical CovR-regulation of genes in mobile elements.** (A) RNA-seq fold changes for each gene were plotted along the BM110 chromosome (x axis). The four genomic regions with an opposite transcription in the Δ*covR* (blue line) and CovR$_{D53A}$ (red line) are highlighted with dotted boxes. (B) Zoom in the four genomic regions highlighted in (A) with genes IDs. (C) Corresponding ChIP-seq profiles obtained after induction of FLAG-CovR with 50 ng/ml in a BM110 Δ*covR* background. Significant (IDR < 0.05) CovR peaks are depicted with black arrows (FE > 4) or grey arrowheads (1 < FE < 4) below the schematic ORF representation of the loci. The CovR binding sites are reported in (B) for comparison with RNA-seq data.
(TIF)

**S3 Fig. *In vitro* phosphorylation and binding of rCovR.** (A) Control of rCovR phosphorylation by acetyl phosphate. Recombinant rCovR was incubated without (-) or with (+) acetyl-phosphate for 1 hour at 37°C. One-half of the sample incubated with acetyl phosphate was subsequently heated at 100°C for 1 minute to remove the labile aspartate phosphorylation. Samples were separated on a 12% Phos-Tag SDS polyacrylamide gel and transferred on membrane before revelation with anti-His antibodies. The Phos-tag gel delays the migration of phosphorylated proteins compared to non-phosphorylated proteins. (B) Negative binding control of rCovR on the P$_{gyrA}$ promoter by EMSA. Increased quantity of purified recombinant protein without (rCovR) or with (rCovR$^P$) phosphorylation by acetyl phosphate are mixed with radiolabeled P$_{gyrA}$ probe and separated on 5% PAA-gel before autoradiography revelation. Related to Fig 3. (C) DNase I protection assay with rCovR on three genomic loci. Single-end radiolabeled probe are mixed with increasing quantity of rCovR before DNaseI treatment. Binding of rCovR on DNA is visualized by the DNA sequences protected from DNase I degradation ('footprints' highlighted by vertical black lines) after separation on Maxam-Gilbert sequencing gels. For the 3 loci, the corresponding rCovR protected sequences are given with the gene IDs in BM110 (BQ8897_RSxxxxx) and NEM316 (GBS_RSxxxxx). The TSSs are indicated by an arrow (+1), the ATG start codon by a box when it is within the sequence, and the SNPs between BM110 and NEM316 by the nucleotides in brackets. Note that different rCovR batches have been used for the three experiments, hampering a direct comparison of rCovR affinities between promoters.
(TIF)

**S4 Fig. CovR transcriptional regulation of cell-wall protein encoding genes.** Histogram of RNA-seq fold changes (Log2 FC) in the CovR$_{D53A}$ mutants for the LPxTG encoding genes identified in the core and accessory genomes of BM110 (red upper panel) and NEM316 (black and white bottom panel). Columns in the NEM316 panel are color-coded following their similarities with BM110 proteins (black: orthologous genes; white: homologous genes including allelic variants and exclusive locus switching; grey: NEM316 specific genes; N/A: no homologue). The LPxTG encoded in integrative and conjugative elements (ICE) are grouped on the left due to the presence of several copies in NEM316 and the presence of several homologous but divergent genes between the two backgrounds. Translation of non-functional proteins due to mutations (frameshifts or internal stop) are indicated by hatched columns in BM110.

Vertical lines on the three PI-1 proteins denoted the loss-of-function mutation (frameshift) of the PI-1 transcriptional activator in NEM316.
(TIF)

**S5 Fig. Comparison of CovR binding on the NEM316 and BM110 chromosomes.** (A) Venn diagrams showing the overlaps between BM110 (orange and red) and NEM316 (blue and light blue) ChIP-seq experiments and between two levels of FLAG-CovR induction (50 and 200 ng/ml anhydro-tetracycline (aTc)). The numbers indicate significant peaks with a Fold Enrichment (FE) superior to 4 that are in common between two experiments or specific to one condition. (B) Linear representation of a consensus sequence between the NEM316 and BM110 chromosomes with the corresponding position of CovR binding sites identified by ChIP-seq. The consensus sequence represents nearly identical regions (yellow), regions with a high frequency of SNPs (green), and strain-specific sequences (white). The binding regions of CovR identified in each ChIP-seq experiment (NEM316 and BM110 at 50 and 200 ng/ml aTc induction) are symbolized by vertical grey lines.
(TIF)

**S1 Table. RNA-seq analysis of *covR* mutants in BM110.** S1A: Whole-genome analysis of RNA-seq in BM110 CovR$_{D53A}$ and Δ*covR*. S1B: BM110 CovR$_{D53A}$ up-regulated genes (Log2 FC > 1, adjusted p-value < 0.005). S1C: BM110 CovR$_{D53A}$ down-regulated genes (Log2 FC < -1, adjusted p-value < 0.005). S1D: BM110 Δ *covR* up-regulated genes (Log2 FC > 1, adjusted p-value < 0.005). S1E: BM110 Δ *covR* down-regulated genes (Log2 FC < -1, adjusted p-value < 0.005). S1F: The BM110 *covR* regulon.
(XLSX)

**S2 Table. ChIP-seq analysis of CovR binding on the BM110 chromosome.** S2A: CovR binding on the BM110 chromosome (with 50 ng/ml aTc, FE > 4 and IDR < 0.05). S2B: Transcriptional changes associated to CovR binding.
(XLSX)

**S3 Table. Transcriptional Start Sites (TSSs) in the BM110 chromosome.**
(XLSX)

**S4 Table. Specificities of the CovR signaling pathway in the hypervirulent CC-17 lineage.**
(XLSX)

**S5 Table. RNA-seq analysis of the CovR$_{D53A}$ mutant in NEM316.** S5A: Whole-genome analysis of RNA-seq in NEM316 CovR$_{D53A}$. S5B: Significantly and differentially transcribed genes (-1 > Log2 FC > 1, adjusted p-value < 0.005) in NEM316 CovR$_{D53A}$.
(XLSX)

**S6 Table. RNA-seq analysis of the CovR core regulon in BM110 and NEM316.** S6A. Whole RNA-seq data on the 1,716 orthologous genes in NEM316 and BM110. S6B. The CovR$_{D53A}$ core regulon. S6C. The CovR$_{D53A}$ strain-specific regulon.
(XLSX)

**S7 Table. LPxTG anchoring motif containing proteins in BM110.**
(XLSX)

**S8 Table. ChIP-seq analysis of CovR binding on the BM110 and NEM316 chromosomes at two levels of CovR induction.** S8A: CovR binding on the NEM316 chromosome with 50 ng/ml aTc and FE > 4. S8B: CovR binding on the NEM316 chromosome with 200 ng/ml aTc and FE > 4. S8C: CovR binding on the BM110 chromosome with 200 ng/ml aTc and

FE > 4.
(XLSX)

**S9 Table. Comparative analysis of covR binding on the BM110 and NEM316 chromosomes.** S9A: Chromosomal CovR binding overlap between BM110 and NEM316. S9B: Sequences similarities between BM110 and NEM316 CovR binding regions (250 bp).
(XLSX)

**S10 Table. IDR analysis of ChIP-seq experiments (FE > 1, IDR < 0.05).** S10A: MACS2 and IDR analysis of ChIP-seq data (BM110 / 0 ng/ml aTc). S10B: MACS2 and IDR analysis of ChIP-seq data (BM110 / 50 ng/ml aTc). S10C: MACS2 and IDR analysis of ChIP-seq data (BM110 / 200 ng/ml aTc). S10D: MACS2 and IDR analysis of ChIP-seq data (NEM316 / 0 ng/ml aTc). S10E: MACS2 and IDR analysis of ChIP-seq data (NEM316 / 50 ng/ml aTc). S10F: MACS2 and IDR analysis of ChIP-seq data (NEM316 / 200 ng/ml aTc).
(XLSX)

**S11 Table. Strains, oligonucleotides, and plasmids.** S11A. Strains used in this study. S11B. Plasmids used in this study. S11C: Oligonucleotides sequences. S11D. Detailed plasmid construction.
(XLSX)

**S12 Table. Genome sequence of *covR* mutants.** S12A. Genome sequence of *covR* mutants in BM110. S12B. Genome sequence of *covR* mutants in NEM316.
(XLSX)

## Acknowledgments

We are thankful to R. Koszul, M. Marbouty, and A. Thierry for their technical assistance in sequencing analysis and for stimulating discussions.

## Author Contributions

**Conceptualization:** Maria-Vittoria Mazzuoli, Arnaud Firon.

**Data curation:** Maria-Vittoria Mazzuoli, Rachel Legendre, Arnaud Firon.

**Formal analysis:** Maria-Vittoria Mazzuoli, Maëlle Daunesse, Hugo Varet, Isabelle Rosinski-Chupin, Rachel Legendre, Philippe Glaser, Claudia Chica, Arnaud Firon.

**Funding acquisition:** Patrick Trieu-Cuot, Arnaud Firon.

**Investigation:** Maria-Vittoria Mazzuoli, Isabelle Rosinski-Chupin, Odile Sismeiro, Myriam Gominet, Pierre Alexandre Kaminski.

**Software:** Maëlle Daunesse, Hugo Varet, Claudia Chica.

**Supervision:** Patrick Trieu-Cuot, Arnaud Firon.

**Writing – original draft:** Maria-Vittoria Mazzuoli, Arnaud Firon.

**Writing – review & editing:** Patrick Trieu-Cuot, Arnaud Firon.

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
