## [Decision Letter · Decision Letter 0]

5 May 2021

Dear Dr Firon,

Thank you very much for submitting your Research Article entitled 'The CovR regulatory network drives the evolution of Group B Streptococcus virulence' to PLOS Genetics.

The manuscript was fully evaluated at the editorial level and by independent peer reviewers. The reviewers appreciated the attention to an important problem, but raised some substantial concerns about the current manuscript. Based on the reviews, we will not be able to accept this version of the manuscript, but we would be willing to review a much-revised version. We cannot, of course, promise publication at that time.

If you decide to revise the manuscript for further consideration at PLOS Genetics, please aim to resubmit within the next 60 days, unless it will take extra time to address the concerns of the reviewers, in which case we would appreciate an expected resubmission date by email to plosgenetics@plos.org.

[LINK]

We are sorry that we cannot be more positive about your manuscript at this stage. Please do not hesitate to contact us if you have any concerns or questions.

Yours sincerely,

Danielle A. Garsin

Associate Editor

PLOS Genetics

Josep Casadesús

Section Editor: Prokaryotic Genetics

PLOS Genetics

Reviewer's Responses to Questions

**Comments to the Authors:**

Reviewer #1: The study by Mazzuoli and colleagues is a useful addition to the large amount of previously-generated data with regard to the importance of the CovR/S regulatory system in the virulence of GBS. This study uses powerful, genome-wide technologies to characterize molecular explanations for the variation in the CovR regulons previously observed between GBS isolates. The combining of RNA-Seq and ChIP-Seq datasets was particularly powerful, and has provided new insights into the specifics of CovR-mediated regulation (e.g. direct vs indirect regulation). For the most part, the experiments performed we done so in scientifically-sound manners, were explained sufficiently such that they could be repeated by others, and the resultant data are presented in engaging and informative manners. While I am excited about the topic as a whole, there are some issues that I believe should be addressed regarding specific aspects of the research:

Major

• Given the highly-sensitive comparisons done with the mutant strains, particularly the transcriptome comparisons between the ΔcovR and covR-D53A mutant strains, it needs to be confirmed that these strains harbor no spurious mutations. This can be achieved by complementing each strain with covR (inserted into the chromosome so that there are no gene-dosage consequences) and repeating the RNA-Seq to show that both complemented strains are identical to the parental (WT) strain. One of several alternatives, and probably an easier approach at this stage, would be for you to perform WGS on the two covR mutants and confirm that the covR mutations are the only ones in the genome that distinguish them from the WT strain. This should also be done with the strains shown in Figure 5.

• In the strains in which FLAG-CovR is induced, what is the concentration of CovR in the cells? How does this compare to CovR levels in the WT strain? The simplest way to look at this would be to use an anti-CovR antibody to do a side-by-side Western. I worry that, due to the placing of CovR on a multicopy plasmid, that the level of CovR is not physiologically-relevant…..which would have a big impact on the global binding characteristics and therefore on the relevance of the ChIP-Seq data (too little, and only the high-affinity promoters will be bound, too much, and off-site binding may occur).

• I see no data that confirms that the addition of the FLAG-tag to CovR does not alter the regulatory activity of the protein. If it does, then this negatively impacts the data gained from the use of this strain. Placing the FLAG-tag into the chromosomally-encoded covR gene in the parental strain (and not via a multi-copy plasmid….due to gene dosage concerns…..which are particularly pronounced and worrisome for regulatory genes) and showing that this strain is identical to the parental strain for the mRNA levels of a range (i.e. some with high-affinity CovR promoters and some low-affinity) of regulated genes would be one way to do this.

• Fig 1B shows hvgA being regulated by CovR with a fold-change log2 value of 7, while Fig S1D shows that this gene only has a fold-change of a log2 value of 2.5-3 following the induction of covR. What accounts for this discrepancy? It should be discussed.

• To significantly strengthen the hypothesis that there are differences in the CovR regulon between CC17 and CC23 strains additional isolates (e.g. 3-5) of each CC should be investigated. This could be, for example, via qRT-PCR of select mRNAs. Otherwise, the described differences between BM110 and NEM316 may simply be strain-specific.

Minor

• There appears to be a discrepancy between figures 1B and 1E. In 1B, it looks like srr2 is differentially-regulated at a higher rate in the ΔcovR mutant than the D53A mutant, but the opposite is true in figure 1E. I believe the labeling is mixed up for figure 1E?

• I don’t know if binding of non-phosphorylated CovR to DNA has been shown in GBS, but I know it has for GAS. Either way, I would add a reference at the end of line 132 that highlights this for GBS (or GAS if no GBS example is known).

• To expand access to this work for people outside of the GBS field, I would add a little more info (1 or 2 sentences) about the bibA/hvgA (line 58) and srr1/srr2 stories (line 59).

• To enhance viewing and interpretation, I suggest adding a faded diagonal line (bottom left to top right) to figures 1B and 5C that would highlight identity between the strains.

• What are the values listed on the Y-axis of the graphs in figure S2B?

Reviewer #2: The CovRS TCS of S. agalactiae (Group B Strep, GBS) is a critical sensory system responsible for expression modulation of roughly 10-15% of the genome, including genes critical in infection. Although the CovRS system has been thoroughly studied in S. pyogenes and several strains of GBS, genome analysis of the CovR regulon has not been described for clonal complex CC-17, which is most highly associated with late-onset meningitis in neonates that is disseminated globally—it is a critical group to be analyzed. Several fundamental questions surrounding the CovRS regulatory system remain unanswered and include determining which genes are under its control in CC-17, and which of these are under CovR’s direct regulation. The present study utilizes RNA-sequencing and ChIP-sequencing studies on two types of CovR loss-of-function mutants (a covR deletion and a D53A point mutant unable to undergo phosphorylation). All experimental methodologies are sound are and rigorously conducted.

Primary findings:

1. The manuscript’s primary source of data presents the genetic regulon of CovR in CC-17 strain BM110 and informs on CovR binding sites across the genome. The effort placed into confirming DNA binding specificity and location is substantial and of extraordinarily high quality. Figures 1-3 and suppl. Figures S1-S3 all provide excellent content.

2. Indirect and secondary regulation by CovR is described thoroughly and with appropriate discussion. Suggestion: possibly consider citing this original observation of N-NS in salmonella (DOI: 10.1126/science.1128794).

3. The manuscript’s most important concept, that CovR-regulated genes are disproportionately under positive selective pressure, is evaluated by genome-wide analysis of mutation frequencies comparing CovR-regulated and unregulated genes, as well as comparisons to a non-CC-17 strain. Bolstering the conclusions that the CovRS regulon is highly plastic and varied between isolates are demonstrations that kinase/phosphatase activities are strikingly different between BM110 and NEM316. Describing differences in the gene regulon, and differences in kinase/phosphatase activities between strains is not necessarily a new concept (especially in literature describing S. pyogenes CovRS regulation); however, the present report make an explicit case that the regulatory system governs the most critical gene sets that account for intra-species evolution and host adaptation. While these conclusions might require a deeper dive into genome comparisons to flesh out rates of variation, it is my sense that the presented data are sufficient to make an important contribution to field.

Other suggestions

1. Fig. 2D. Identification of the binding site is interesting, important, and matches very well to what has been documented in S.pyogenes. However, there has been some controversy over this in the pyogens literature (please see https://doi.org/10.1046/j.1365-2958.2002.02810.x compared to https://doi.org/10.1038/srep12057). Perhaps this should be mentioned in the results or discussion.

2. Line 142. Should state explicitly what is represented by ‘peaks’ (sites where CovR binds).

3. Line 201. It is not clear what is meant by, “the functions encode by groups 1 and 2 genes did not differ from the function encode by genes of the direct regulon.”

4. Line 233, it was unclear to me which categories of figure 4 that the “24 genes associated with CovR binding accumulated more mutations than expected…”

5. Is anything known about what CovS responds to? Again, given the close parallels, I think it would benefit a reader to know that in S. pyogenes that LL-37 and Mg2+ are modulators of CovS kinase/phosphatase activity. Perhaps the paragraph at line 396 would be an ideal place to discuss this.

**Have all data underlying the figures and results presented in the manuscript been provided?**

Reviewer #1: Yes

Reviewer #2: Yes

PLOS authors have the option to publish the peer review history of their article (what does this mean?). If published, this will include your full peer review and any attached files.

Reviewer #1: No

Reviewer #2: No

---

## [Decision Letter · Decision Letter 1]

9 Aug 2021

Dear Dr Firon,

We are pleased to inform you that your manuscript entitled "The CovR regulatory network drives the evolution of Group B Streptococcus virulence" has been editorially accepted for publication in PLOS Genetics. Congratulations!

Yours sincerely,

Danielle A. Garsin

Associate Editor

PLOS Genetics

Josep Casadesús

Section Editor: Prokaryotic Genetics

PLOS Genetics

Comments from the reviewers (if applicable):

Reviewer's Responses to Questions

**Comments to the Authors:**

Reviewer #1: The revised manuscript by Mazzuoli et al. describes an impactful study of CovR/S function in the group A Streptococcus. The authors did a commendable job in replying to the reviewer comments that were made after the initial submission, including performing multiple new experiments that has solidified the findings made in the initial submission. I have no concerns with the revised version.

Reviewer #2: I have no further concerns or suggestions for the authors.

**Have all data underlying the figures and results presented in the manuscript been provided?**

Reviewer #1: Yes

Reviewer #2: Yes

PLOS authors have the option to publish the peer review history of their article (what does this mean?). If published, this will include your full peer review and any attached files.

Reviewer #1: No

Reviewer #2: No

**Data Deposition**

http://datadryad.org/submit?journalID=pgenetics&manu=PGENETICS-D-21-00538R1

**Press Queries**

---

## [Editor Report · Acceptance letter]

2 Sep 2021

PGENETICS-D-21-00538R1 

The CovR regulatory network drives the evolution of Group B Streptococcus virulence 

Dear Dr Firon, 

We are pleased to inform you that your manuscript entitled "The CovR regulatory network drives the evolution of Group B Streptococcus virulence" has been formally accepted for publication in PLOS Genetics! Your manuscript is now with our production department and you will be notified of the publication date in due course.

With kind regards,

Agnes Pap

PLOS Genetics

On behalf of:
